# Toxic Effects on Oxidative Stress, Neurotoxicity, Stress, and Immune Responses in Juvenile Olive Flounder, *Paralichthys olivaceus,* Exposed to Waterborne Hexavalent Chromium

**DOI:** 10.3390/biology11050766

**Published:** 2022-05-17

**Authors:** Ju-Wook Lee, Jun-Hwan Kim, Deok-Chan Lee, Hyun-Jeong Lim, Ju-Chan Kang

**Affiliations:** 1West Sea Fisheries Research Institute, National Institute of Fisheries Science, Incheon 22383, Korea; leejuwook84@gmail.com; 2Department of Aquatic Life Medicine, Pukyong National University, Busan 48513, Korea; 3Department of Aquatic Life and Medical Science, Sun Moon University, Asan 31460, Korea; 4Aquaculture Industry Research Division, South Sea Fisheries Research Institute, National Institute of Fisheries Science, Yeosu 59780, Korea; saranghase@korea.kr (D.-C.L.); limhj@korea.kr (H.-J.L.)

**Keywords:** hexavalent chromium, antioxidant responses, neurotransmitter, stress responses, immune responses

## Abstract

**Simple Summary:**

Metals such as chromium can be exposed at high levels in the marine environment, and exposure to these heavy metals can have a direct effect on marine organisms. High levels of chromium exposure can have a direct impact on organisms in a coastal cage and terrestrial aquaculture. Hexavalent chromium exposure of more than 1.0 and 2.0 mg Cr^6+^/L induced physiological responses such as antioxidant, neurotransmitter, immune, and stress indicators in *Paralichthys olivaceus*. Therefore, this study will provide a reference indicator for stable aquaculture production through reference indicators for toxicity due to chromium exposure that may exist in the marine environment.

**Abstract:**

Juvenile *P**aralichthys olivaceus* were exposed to waterborne hexavalent chromium at various concentrations (0, 0.5, 1.0, and 2.0 mg/L) for 10 days. After chromium exposure, the activities of superoxide dismutase and glutathione S-transferase, which are oxidative stress indicators, were significantly increased; however, the glutathione level was significantly reduced. Acetylcholinesterase activity as a neurotoxicity marker was significantly inhibited upon chromium exposure. Other stress indicators, including plasma cortisol and heat shock protein 70, were significantly increased. The immune response markers (lysozyme and immunoglobulin M) were significantly decreased after chromium exposure. These results suggest that exposure to environmental toxicity in the form of waterborne chromium at concentrations higher than 1.0 mg/L causes significant alterations in antioxidant responses, neurotransmitters, stress, and immune responses in juvenile olive flounders. This study will provide a basis for an accurate assessment of the toxic effects of hexavalent chromium on aquatic organisms.

## 1. Introduction

In aquatic environments, high levels of metal contamination from industrial activities can induce metal toxicity in aquatic animals. Hexavalent chromium is one of the most toxic metals to aquatic animals, as it readily penetrates cell membranes [1]. In water, hexavalent chromium is mainly present as the chromate ion and has 500–1000 times higher activity than trivalent chromium, causing higher toxicity which can result in allergic reactions, deformities, tissue damage, and abnormal behavior in fish [2,3]. Hexavalent chromium does not readily degrade naturally and thus can accumulate in the environment and have direct and indirect effects on aquatic animals, especially fish [4,5]. Inside cells, hexavalent chromium is reduced to trivalent chromium, which forms complexes with intracellular macromolecules, including genetic material, causing toxicity and mutagenicity.

Exposure to metals, including hexavalent chromium, induces the overproduction of reactive oxygen species (ROS), including the superoxide radical, hydroxyl radical, and hydrogen peroxide [6]. In particular, hexavalent chromium is related to ROS activation as it is reduced by cellular reductants to trivalent chromium via the reactive intermediates pentavalent chromium and tetravalent chromium, thereby catalyzing a Fenton-like redox-cycling reaction [7,8,9]. Fish have developed an antioxidant system to protect themselves from oxidative stress by removing free radicals. Superoxide dismutase (SOD), glutathione S-transferase (GST), and glutathione (GSH) are the major antioxidant responses molecules in fish and are widely used to indicate and evaluate oxidative stress in aquatic animals [10]. In fish, oxidative stress generally incudes two cellular processes: injury of cellular components and stimulation of antioxidant defense systems [11]. The analysis of such antioxidant changes would allow the assessment of oxidative stress caused by waterborne hexavalent chromium exposure in fish [12,13].

Excess ROS-related oxidative stress can induce cell death, including apoptosis in fish, and result in neurotoxicity [14]. In particular, hexavalent chromium exposure causes apoptosis in fish livers and induces neurotoxicity [15]. The neurotransmitter acetylcholinesterase (AChE) participates in cognitive processes, and acts via muscarinic and cholinergic receptors [16]. AChE also plays a critical role in the central and peripheral nervous systems, where it breaks down acetylcholine into choline and acetic acid, which then diffuse in the nervous synapses [17]. Thus, AChE maintains appropriate levels of acetylcholine, and AChE inhibition leads to the accumulation of acetylcholine at the synaptic junctions, which is a characteristic feature of neurotoxicity [18]. Since AChE is an important marker for evaluating neurotoxicity, and chromium toxicity affects neurotransmitters, AChE may be a reliable indicator for assessing neurotoxicity induced by hexavalent chromium exposure in fish.

Exposure to metals, including hexavalent chromium, impairs fish kidney and liver functions and induces stress [19]. In teleost fish, cortisol, a corticosteroid, is secreted by interrenal cells present in the head kidney and is widely used as an indicator of physiological stress and health status of fish inhabiting toxic environments [20,21]. An increase in cortisol secretion in fish is considered the main adaptive response to acute stress by maintain osmotic homeostasis and the gluconeogenic pathway [22]. Heat shock proteins (HSPs) are biochemical biomarkers induced by various physical, chemical, and biological stressors such as temperature changes, tissue trauma, infection, and metal toxicity, and are widely used to evaluate environmental toxicity [23]. HSP-70 is a major component of protein folding and transport and acts by binding target proteins. Under stress, HSP-70 is usually activated as part of the detoxification mechanism to repair cells under stress [24]. Thus, determining the stress response can be useful to evaluate toxic effects in fish exposed to hexavalent chromium.

Toxicity induced by metal exposure, particularly hexavalent chromium, affects fish immunity by inducing physiological alterations. Prabakaran et al. [25] suggested that immune toxicity induced by hexavalent chromium affects the non-specific immunity and the overall immune response in fish. Lysozyme is a major anti-microbial protein that breaks the β-1,4 glycosidic bond between N-acetylmuramic acid and N-acetylglucosamine in the cell wall of Gram-positive bacteria [26]. In fish, lysozyme activity is affected by aquatic pollution, and is widely used to evaluate toxic effects of various metals [17]. Plasma immunoglobulin is a major component of the teleost humoral immune system; immunoglobulin M (Ig-M) is the most important immunoglobulin to evaluate health status and stress in fish exposed to toxic environment [27]. Environmental toxins can induce immune suppression or enhance immune response [28], and these immunity indicators are important for evaluating toxicity induced by metal exposure.

Olive flounder (*Paralichthys olivaceus*) inhabits the coastlines of Korea and is the most commercially important aquaculture product in Korea owing to high consumer demand. Hexavalent chromium can be present at relatively high levels in the ocean near the coastline due to industrial activities and can exert toxic effects on aquatic animals. However, there are insufficient scientific data delineating the toxic effects of hexavalent chromium on live flounder. In general, concentrations of hexavalent chromium in the marine environment are low (5–800 μg/L) [29,30]. However, hexavalent chromium has been continuously released through anthropogenic activities such as industrial facilities, and concentrations of chromium up to 2.4 mg/L were detected in the coastal industrial area of Yangchen, China [30]. These pollutants are detected at higher than environmental concentrations in organisms with high nutrient levels organisms such as fish because of biological enrichment effects. In addition, previous studies have used hexavalent chromium at concentrations (0.05–41.75 mg/L) higher than the average of seawater to study the clear biological effects [30,31,32]. Hexavalent chromium is absorbed directly from the gills and accumulates in organs such as the liver and kidneys. Chromium accumulates significantly more in the liver and gills than in the muscles and intestines of fish, and the accumulation tends to increase as the concentration of chromium increases [33]. Therefore, we aimed to evaluate hexavalent-chromium-mediated oxidative stress, neurotoxicity, stress, and immune responses in the liver and gill of *P. olivaceus* following its exposure to hexavalent chromium and identify an indicator of hexavalent chromium toxicity in this species.

## 2. Materials and Methods

### 2.1. Experimental Fish and Conditions

Juvenile olive flounder (*Paralichthys olivaceus*) were exposed to various waterborne hexavalent chromium at 0, 0.5, 1.0, and 2.0 mg/L. The fish were acclimatized for a month under laboratory conditions before hexavalent exposure, and the seawater components used in the experiment are listed in Table 1. After acclimatization, 48 fishes (mean weight: 25.6 ± 3.7 g, mean length: 13.3 ± 1.6 cm) were selected for the study, and chromium exposure took place in 100 L circular tanks containing 6 fish per treatment group in duplicates. Hexavalent chromium (potassium dichromate, K_2_Cr_2_O_7_, Sigma Chemical, St. Louis, MO, USA, ≥99.0%) was dissolved in distilled water to make a standard stock solution of 1000 mg/L, and the concentration of 0, 0.5, 1.0, and 2.0 mg/L in each water tank. The circular tank water was thoroughly exchanged on alternate days, and the required concentration was again maintained in respective tanks. At the end of the culture at 5 and 10 days, fish were anesthetized in buffered 3-aminobenzoic acid ethyl ester methanesulfonate (Sigma Chemical, St. Louis, MO, USA). The liver and gill of the sample were analyzed, and the actual level of hexavalent chromium in each exposure experiment tank was measured is shown in Table 2. The seawaters were filtered through a 0.2 µm membrane filter (Advantec MFS, Inc., Dublin, CA, USA) under pressure for analysis. The seawaters were analyzed by the Elan DRC ICP-MS instrument with argon gas (ELAN 6600DRC, Perkin-Elmer, Shelton, CT, USA) to determine total chromium concentrations. ICP multi-element standard solution VI (Merck, Darmstadt, Germany) was used for a standard curve. Total chromium concentrations in the specimens were determined by external calibration. Although the exposure concentration of chromium used in this study is higher than the level existing in the actual marine environment, it is possible to accumulate an indicator of toxicity standard of fish due to chromium exposure through exposure experiments at this concentration.

### 2.2. Antioxidant Responses

Liver and gill tissues were sampled, diluted 10 times using 0.1 M PBS buffer, and homogenized to confirm the antioxidant reaction. After homogenizing the samples, the samples were centrifuged at 10,000× *g* rpm for 30 min using a 4 °C centrifuge (Smart-R17, Hanil Science Industrial, Gimpo, Korea), and the supernatant was separated and used for analysis. Superoxide dismutase (SOD) activity was measured with a 50% inhibitor rate about the reduction reaction of WST-1 using the SOD Assay kit (Dojindo Molecular Technologies, Inc., Rockville, MD, USA). One unit of SOD is defined as the amount of the enzyme in the 20 μL of sample solution that inhibits the reduction reaction of WST-1 with superoxide anion by 50%. SOD activity was shown as unit mg protein^−1^.

Glutathione-S-transferase (GST) activity was measured according to the method of Kim et al. [34]. The reaction mixture composed of 0.2 M phosphate buffer (pH 6.5), 10 mM GSH (Sigma) and 10 mM 1-chloro-2,-dinitrobenzene, CDNB (Sigma). The change in absorbance at 25 °C was measured at 340 nm and the enzyme activity was expressed as nmol min^−1^ mg protein^−1^. Reduced glutathione (GSH) was followed the method of Kim and Kang [6]. Briefly, 0.2 mL of fresh sample supernatant was added to 1.8 mL of distilled water. Three mL of the precipitating solution (1.67 g metaphosphoric acid, 0.2 g EDTA and 30 g NaCl in 100 mL distilled water) was mixed with supernatants. The mixture was centrifuged at 10,000× *g* rpm for 10 min (Smart-R17, Hanil Science Industry, Gimpo, Korea). One mL of supernatant was added to 4.0 mL of 0.3 M NaHPO_4_ solution. Furthermore, 0.5 mL 5,5′-dithiobis-2-nitrobenzoic acid (DTNB) was then added to this solution. GSH was measured as the difference in the absorbance values at 412 nm between samples in the presence and the absence of DTNB, and the value was expressed as μmol mg protein^−1^ in the tissues.

### 2.3. Acetylcholinesterase Activity

To confirm AChE activity, liver and gill tissues were sampled, homogenized using 0.1 M PBS buffer (10 times), and the supernatant was separated at 10,000 rpm for 30 min using a 4 °C centrifuge (Smart-R17, Hanil Science Industry, Gimpo, Korea). AChE activity was normalized to protein content and expressed as nmol min^−1^ mg protein^−1^. The activity of the homogenate was measured by determining the rate of hydrolysis of acetylthiocholine iodide (ACSCh, 0.88 mM) in a final volume of 300 μL, with 33 μL of 0.1 M phosphate buffer, pH 7.5 and 2 mM DTNB. The reaction was started with the addition of the substrate acetylthiocholine. As soon as the substrate was added, the hydrolysis and the formation of the dianion of DTNB were analyzed at 412 nm for 5 min using a microplate reader in intervals of 1 min.

### 2.4. Stress Indicators

Plasma cortisol and HSP-70 indicators were used to confirm the stress indicators. Cortisol was confirmed by separating plasma from blood samples through centrifugation. Plasma cortisol was measured using a monoclonal antibody enzyme-linked immunosorbent assay (ELISA) quantification kit (Enzo Life Sciences, Inc., Farmingdale, NY, USA). According to the kit manual, cortisol standards and samples were measured using an anti-Mouse Ig G microtiter plate. After all this, optical density was measured at 450 nm. The measurements were performed in triplicate.

Liver and gill homogenized samples were used to determine HSP-70 in the liver and gills, and analysis was carried out using the ELISA assay kit (MyBioSource, Inc., San Diego, CA, USA) [35]. According to the kit manual, HSP 70 standards and samples were measured. After all this, absorbance value was analyzed at 450 nm using a reader within 15 min after adding the stop solution.

### 2.5. Immune Responses

Lysozyme activity was analyzed using plasma samples, and analysis was performed according to the method of Kim et al. [36]. Plasma Ig-M concentrations were measured using an enzyme-linked immunosorbent assay (ELISA) quantification kit (MyBioSource Inc., San Diego, CA, USA) [37]. Ig-M standards and plasma samples were measured as manual methods. The measurements were carried out in triplicate. After the process, the absorbance value was determined at 450 nm within 15 min.

### 2.6. Statistical Analysis

Statistical analyses were carried out using the SPSS/PC+ statistical package (SPSS Inc., Chicago, IL, USA). Significant differences between results were identified using one-way analysis of variance (ANOVA) and Tukey’s test for multiple comparisons. The significance level was set at *p* < 0.05.

## 3. Results

### 3.1. Antioxidant Responses

Antioxidant responses of olive flounder (*P. olivaceus*) exposed to waterborne hexavalent chromium are shown in Figure 1. The SOD activity in the liver was significantly increased over 1.0 mg/L both in 5 and 10 days by chromium exposure (*p* < 0.05). A significant increase in the gill SOD activity was observed at 2.0 mg/L in 5 days and over 1.0 mg/L in 10 days (*p* < 0.05). The GST activity in the liver was significantly increased over 1.0 mg/L in 5 days and over 0.5 mg/L in 10 days (*p* < 0.05). In the gill tissues, the GST activity was significantly increased at 2.0 mg/L both in 5 and 10 days (*p* < 0.05). The GSH level in the liver was significantly decreased over 1.0 mg/L in 5 days and over 0.5 mg/L in 10 days (*p* < 0.05). The GSH level in the gill was also significantly decreased at 2.0 mg/L in 5 days and over 1.0 mg/L in 10 days (*p* < 0.05).

### 3.2. Acetylcholinesterase Activity

AChE in *P. olivaceus* exposed to waterborne hexavalent chromium is shown in Figure 2. The AChE activity in the liver was significantly inhibited at 2.0 mg/L in 5 days and over 1.0 mg/L in 10 days (*p* < 0.05). In the gill tissues, there was no significant change in the AChE in *P. olivaceus* by waterborne hexavalent chromium exposure.

### 3.3. Stress Responses

Plasma cortisol in *P. olivaceus* exposed to waterborne hexavalent chromium is shown in Figure 3. The plasma cortisol in *P. olivaceus* was significantly increased over 1.0 mg/L both in 5 and 10 days by waterborne hexavalent chromium exposure (*p* < 0.05).

Heat shock protein 70 in the liver and gills in *P. olivaceus* exposed to waterborne hexavalent chromium are shown in Figure 4. The HSP-70 in the liver was significantly increased at 2.0 mg/L in 5 days and over 1.0 mg/L in 10 days (*p* < 0.05). In the gill tissues, the HSP-70 was significantly increased over 1.0 mg/L in 10 days (*p* < 0.05), whereas there was no significant change in 5 days.

### 3.4. Immune Responses

Immune responses such as plasma lysozyme and immunoglobulin M in *P. olivaceus* exposed to waterborne hexavalent chromium are shown in Figure 5. The lysozyme activity in the plasma in *P. olivaceus* was significantly decreased at 2.0 mg/L in 5 days and over 1.0 mg/L in 10 days (*p* < 0.05). The Ig-M in the plasma was also significantly decreased over 1.0 mg/L both in 5 and 10 days by waterborne hexavalent chromium exposure (*p* < 0.05).

## 4. Discussion

Hexavalent chromium exposure induces ROS production, which negatively impacts cellular function and integrity by damaging proteins, nucleic acids, and lipids [1,28,38]. Antioxidant responses are activated to cope with the oxidative stress induced by excessive free radical production in response to oxidative stressors, including metal exposure [34]. Among the components of antioxidant response, SOD is a critical first-defense enzyme that converts hydroxyl radicals into hydrogen peroxide [17]. In this study, SOD activity in the liver and gills of *P. olivaceus* increased significantly following hexavalent chromium exposure, most likely as a defense mechanism to convert the excess oxygen and free radicals induced by hexavalent chromium exposure into hydrogen peroxide. Li et al. [1] reported a significant increase in SOD activity in the brain and liver of rainbow trout (*Oncorhynchus mykiss*) that were exposed to hexavalent chromium and suggested that the SOD activation implies an increase in the hydrogen peroxide levels in cells following ROS conversion. Similarly, Chaâbane et al. [35] reported a significant increase in SOD activity in the gills and digestive glands of the bivalve mollusk (*Venus verrucose*) after hexavalent chromium exposure, suggesting that increased SOD activity is a defense mechanism to prevent tissue damage by eliminating excess ROS induced by the hexavalent chromium.

GST is another critical enzyme for free radical and xenobiotic detoxification [39]. It promotes xenobiotic detoxification by conjugating electrophilic metabolites to GSH [40]. Ni et al. [41] reported that hexavalent chromium induced a significant increase in GST activity in marine medaka (*Oryzias melastigma*) and suggested that this represented a mechanism to detoxify the oxidative stress caused by hexavalent chromium. Chaâbane et al. [35] also reported a significant increase in GST activity in the bivalve mollusk (*V. verrucose*) upon hexavalent chromium exposure, suggesting the activation of glutathione–xenobiotic conjugation for excretion. Similarly, Kim and Kang [6] reported a significant increase in GST activity in *Sebastes schlegelii* exposed to hexavalent chromium. In our study, GST activity in *P. olivaceus* increased significantly upon exposure to hexavalent chromium, possibly to reduce the toxicity caused by chromium exposure.

In fish, GSH plays a crucial role in removing free radicals produced in response to metal exposure and in cellular metabolism [42], and the GSH level is a widely used biomarker to determine the cellular redox state [43]. According to Yao et al. [44], hexavalent chromium can generate free radicals in cells by reacting with molecules such as GSH or stimulating NADPH oxidase activity during the reduction process. Thus, glutathione is actively involved in ROS metabolism and can be an appropriate marker for evaluating the chromium-exposure-induced redox status in fish [45]. Chen et al. [31] reported a significant depletion of GSH in Japanese medaka (*Oryzias latipes*) following hexavalent chromium exposure and they suggested that the decrease in GSH was due to an increased activation of enzymes that catalyze the reduction of hydrogen peroxide. Similarly, Yuan et al. [46] reported a significant decrease in the GSH level after an initial increase in GSH upon hexavalent chromium exposure. In the present study, the GSH levels in the liver and gills of *P. olivaceus* were significantly depleted following chromium exposure which is attributed to the oxidation of GSH to GSSG during free radical scavenging, while hexavalent chromium is reduced to trivalent chromium. GSSG, in turn, is recycled to GSH through glutathione reductase in a process using NADPH. We therefore suggest that the GSH depletion observed in this study was due to an excessive production of free radicals following chromium exposure.

Hexavalent chromium acts as a neurotoxin in fish because the intermediate is generated during the conversion of hexavalent chromium to trivalent chromium by reducing agents such as GSH [47]. Hexavalent chromium affects the central nervous system by causing oxidative or nitrosative stress [48,49]. AChE terminates cholinergic reaction nicotinic acetylcholine receptors [50,51]. A typical feature of metal exposure in fish is acetylcholine inhibition which causes acetylcholine accumulation at synaptic junctions, leading to acute cholinergic syndrome and eventually death [52]. Mahmoud and Abd El-Twab [53] reported that hexavalent chromium affects AChE activity because they observed a significant inhibition of AChE in rats exposed to hexavalent chromium. Similarly, Domingues et al. [54] reported a significant inhibition of AChE in zebrafish (*Danio rerio*) exposed to hexavalent chromium. Kim and Kang [6] also reported a significant inhibition of AChE in *S. schlegelii* following hexavalent chromium exposure and they inferred that the AChE dysregulation was induced by severe injury to the nervous tissues through excessive ROS production. Ciacci et al. [55] suggested that hexavalent chromium affects components involved in neurotransmission in aquatic animals. They reported a significant decrease in AChE activity in the Mediterranean mussel (*Mytilus galloprovincialis*)*,* exposed to hexavalent chromium, indicating chromium-toxicity-induced damages to cholinergic signaling. In the present study, AChE was inhibited in the liver of *P. olivaceus* following hexavalent chromium exposure, suggesting that chromium acts as a neurotoxic agent and damage neurotransmitters. Gill activity is regulated by the sympathetic and parasympathetic nerves of the autonomic nervous system distributed through the branch connective tissue, and neurotransmitters regulate the ciliary rhythms [55]. However, in the present study, no significant change was observed in the AChE activity in the gills of *P. olivaceus* following hexavalent chromium exposure, indicating that the neurotoxic effects of chromium may be tissue-dependent. Chromium accumulations in fish vary by tissue type [56]. It accumulates more in the liver than in the gills and muscles of fish, a higher toxic effect is induced in the liver [57]. Therefore, our results indicate that the effects of chromium exposure on neurotoxicity may be limited to specific tissue types.

Hexavalent chromium is as a major stressor in aquatic animals and stress indicators can be reliable and sensitive biomarkers to evaluate toxic effects on fish exposed to hexavalent chromium [58]. Stress responses are commonly induced in fish by various environmental toxicities but differ depending on the fish species, size, and environmental factors as well as on the pollutant dose and exposure time [59]. Increased cortisol secretion is a major stress response in fish following hexavalent chromium exposure, and several studies have suggested a correlation between hexavalent chromium exposure and cortisol secretion [27,60]. According to Kim and Kang [27], cortisol production is generally stimulated in fish to maintain homeostasis by activating the central nervous system and increasing blood pressure. Moreover, hexavalent chromium exposure causes a significant increase in plasma cortisol levels in the rockfish (*S. schlegelii*), which represents a critical stress reaction to activate metabolic processes in response to chromium toxicity. Mishra and Mohanty [60] reported a significant increase in the serum cortisol level in the spotted snakehead (*Channa punctatus*) upon hexavalent chromium exposure and suggested that this increase served as a defense mechanism to chromium toxicity by maintaining osmotic homeostasis and physiological activities. In our study, the plasma cortisol level in *P. olivaceus* was significantly upregulated and the increased cortisol secretion upon acute exposure to chromium may have served to restore homeostasis to protect the fish from physiological disturbances caused by chromium toxicity.

HSPs are essential protein synthesizers and HSP-70 is mainly increased after exposure to oxidative stress as a mechanism to protect cells from ROS damage [61]. HSP-70 is stimulated by environmental stressors and is a reliable biomarker to assess exposure to various metal [62]. Rudolf and Cervinka [15] reported that an increase in the HSP-70 level is closely related to the stress response induced by chromium exposure and that this protein is involved in reactions that interact with and inhibit intrinsic and external apoptotic pathways at multiple sites. Kim et al. [50] reported a significant increase in HSP-70 gene expression in *S. schlegelii* exposed to hexavalent chromium. Padmini and Tharani [63] reported a significant upregulation of HSP-70 in the common mullet (*Mugil cephalus*) under metal-induced stress and suggested that this upregulation may mediate signaling pathways that facilitate cellular resistance to apoptosis. Similarly, Rajeshkumar and Munuswamy [64] reported a significant increase in the HSP-70 level in milk fish (*Chanos chanos*) inhabiting polluted coasts, which provides protection against pollution stress-induced damage. In this study, the HSP-70 level in *P. olivaceus* was significantly increased following exposure to hexavalent chromium, and this increase could be a stress response to chromium exposure and may exert a protective effect against excess free radical generation.

The fish immune system is affected by environmental challenges and chromium exposure is a well-known inducer of immune toxicity [65]. Immune molecules are reliable indicators of the health status of fish under various environmental stressors since acute or chronic exposure to environmental toxicity adversely affects the innate and adaptive immune systems [37,66]. Among the various non-specific immune responses, lysozyme activity is a major immune response indicator in fish that can be used to monitor metal toxicity in fish because it reacts strongly to environmental toxins [59]. According to Borgia et al. [66], metal exposure below the lethal dose can suppress non-specific and specific immunity in fish and lysozyme is suppressed in common carp (*Cyprinus carpio*) exposed to effluent from the electroplating industry. Similarly, hexavalent chromium exposure caused a significant decrease in lysozyme activity in the bivalve mollusk (*Mytilus galloprovincialis*). The authors speculated that this was due to a decrease in hemocytes and related to the suppression of phagocytic activity, caused by the inhibition of enzymatic activity, inhibition of lysozyme activity by chromium-induced immune toxicity [55]. Prabakaran et al. [67] reported that hexavalent chromium exposure at lethal concentrations reduced lysozyme activity in Mozambique tilapia (*Oreochromis mossambicus*), and negatively impacted their resistance to diseases. In this study, lysozyme activity in *P. olivaceus* was significantly inhibited upon chromium exposure, which is believed to represent a decrease in immune capacity due to immune toxicity caused by chromium toxicity.

Immunoglobulins are mostly generated by plasma blasts and plasma cells, and they are secreted into body fluids, including serum, and mucosal secretions as antibodies (i.e., the soluble form) or presented on the B cell surface as B cell receptors (i.e., the membrane-bound form) [68]. Unlike Ig-D, Ig-T, and Ig-Z, which have only been recognized since 1997, Ig-M was first discovered several decades ago. It is denatured by various environmental changes and exists in different redox forms [59]. The Ig-M concentration in teleosts varies depending on the species, size, ambient temperature, water quality, and presence of stressor, and is used as an indicator of environmental toxicity [69]. Metal exposure in fish also acts as a potent antigen and can lead to the unwarranted suppression or stimulation of plasma Ig-M levels [70]. Ren et al. [71] reported significant decrease in plasma Ig-M levels in *P. olivaceus* upon methylmercury exposure and they suggested that apoptosis of B lymphocytes induced by methylmercury toxicity may have been responsible for this decrease. In contrast, exposure to certain metals, such as arsenic and lead induces significant increases in plasma Ig-M levels in *S. schlegelii*, which is believed to represent an immune stimulus for metal detoxification, as metals are recognized as a foreign substance [27,59]. In the tiger puffer (*Takifugu rubripes*), nitrate as well as metal exposure induced significant decreases in the plasma Ig-M level, which indicated immune suppression due to immune toxicity [72]. Albano [73] suggested that the immune suppression may be a consequence of liver damage caused by persistent oxidative stress. In this study, hexavalent chromium exposure induced a significant decrease in the plasma Ig-M levels in *P. olivaceus*, indicating a compromised immune function due to chromium toxicity.

## 5. Conclusions

In conclusion, the results of this study showed that waterborne hexavalent chromium exposure induces significant changes in the antioxidant responses (SOD, GST, and GSH levels and activities) in *P. olivaceus*. In addition, chromium exposure inhibits AChE. Chromium exposure is a stressor and induces significant increases in plasma cortisol and HSP-70 levels in *P. olivaceus*. Furthermore, it leads to lysozyme inhibition and decreases the Ig-M levels in *P. olivaceus*. Hexavalent chromium at a concentration higher than 1 mg/L induces oxidative stress, neurotoxicity, and immune toxicity in *P. olivaceus*. Thus, the physiological changes in *P. olivaceus* can be used as a standard indicator to accurately assess the toxic effects of hexavalent chromium.

## Figures and Tables

**Figure 1 biology-11-00766-f001:**
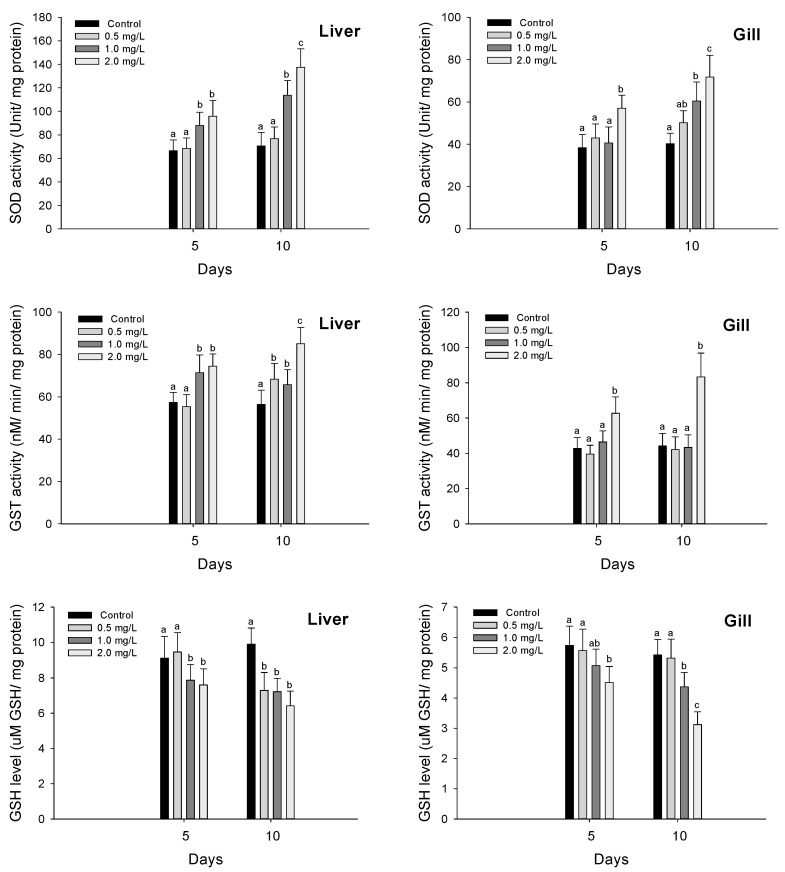
Antioxidant responses of flatfish, *Paralichthys olivaceus* exposed to the different concentrations of waterborne chromium for 10 days. Values with different superscripts are significantly different in 5 and 10 days (*p* < 0.05) as determined by Tukey’s multiple range test.

**Figure 2 biology-11-00766-f002:**
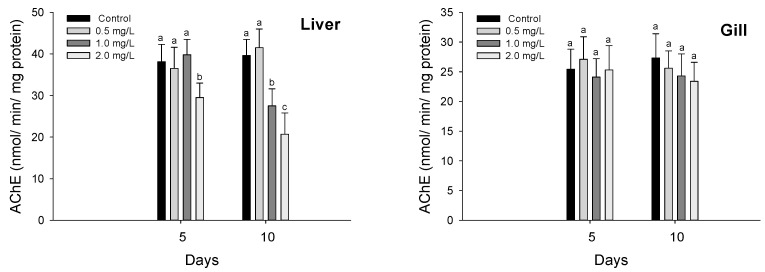
AChE inhibition of flatfish, *Paralichthys olivaceus* exposed to the different concentrations of waterborne chromium for 10 days. Values with different superscripts are significantly different in 5 and 10 days (*p* < 0.05) as determined by Tukey’s multiple range test.

**Figure 3 biology-11-00766-f003:**
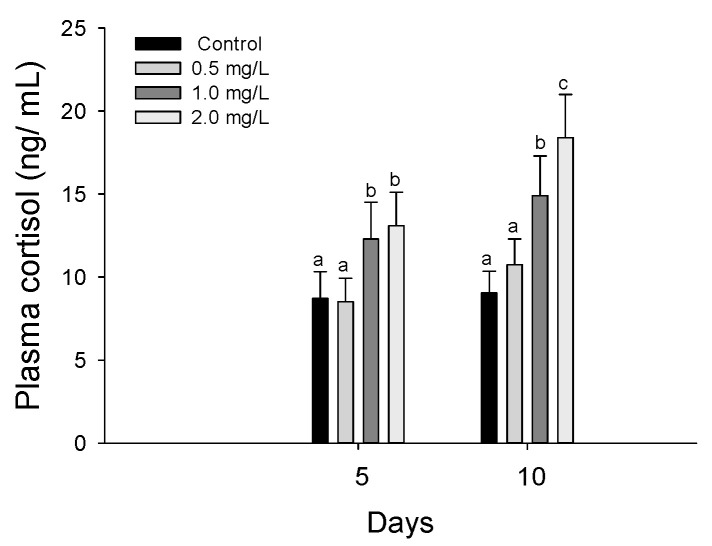
Plasma cortisol of flatfish, *Paralichthys olivaceus,* exposed to the different concentrations of waterborne chromium for 10 days. Values with different superscripts are significantly different in 5 and 10 days (*p* < 0.05) as determined by Tukey’s multiple range test.

**Figure 4 biology-11-00766-f004:**
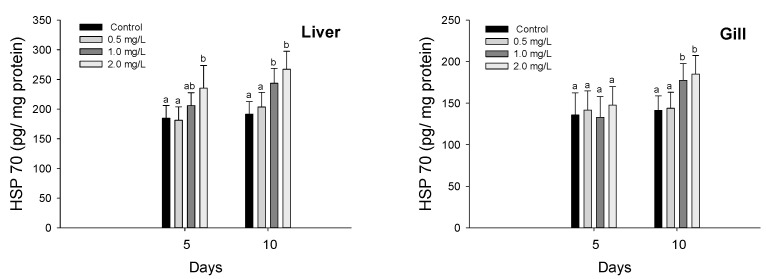
HSP 70 of flatfish, *Paralichthys olivaceus,* exposed to the different concentrations of waterborne chromium for 10 days. Values with different superscripts are significantly different in 5 and 10 days (*p* < 0.05) as determined by Tukey’s multiple range test.

**Figure 5 biology-11-00766-f005:**
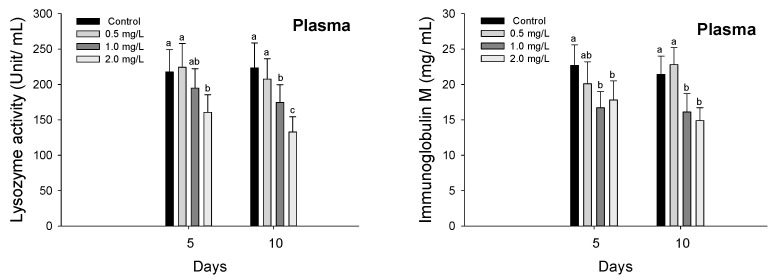
Immune responses of flatfish, *Paralichthys olivaceus* exposed to the different concentrations of waterborne chromium for 10 days. Values with different superscripts are significantly different in 5 and 10 days (*p* < 0.05) as determined by Tukey’s multiple range test.

**Table 1 biology-11-00766-t001:** The chemical components of seawater and experimental condition used in the experiments.

Item	Value
Temperature (°C)	20.0 ± 0.5
pH	7.9 ± 0.2
Salinity (‰)	33.1 ± 0.2
Dissolved Oxygen (mg/L)	8.1 ± 0.4
Ammonia (mg/L)	0.12 ± 0.03
Nitrite (mg/L)	0.16 ± 0.07
Nitrate (mg/L)	1.31 ± 0.24

**Table 2 biology-11-00766-t002:** Comparison of target concentration and the actual concentration of each source.

Waterborne Cr Concentration (mg/L)				
Waterborne Cr concentrations	Control	0.5	1.0	2.0
Measured Cr concentrations	0.01	0.47	1.08	2.13

## Data Availability

Not applicable.

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
