# Peer review of "Toxic Effects on Oxidative Stress, Neurotoxicity, Stress, and Immune Responses in Juvenile Olive Flounder, Paralichthys olivaceus, Exposed to Waterborne Hexavalent Chromium"

_biology, 2022, doi:10.3390/biology11050766_

Round 1

Reviewer 1 Report

Requested changes have been attempted.

Author Response

Dear, reviewer
Thank you for your attentive review.

Reviewer 2 Report

 This revised manuscript has been improved and most of the concerns have been addressed. However, the authors may consider some other issues as bellows.

  1. Description in line 121-124 that explain why the gill and liver was sampled should be removed to Introduction. Similarly, Description in line 131-138 also should be removed to Introduction. In Materials and Methods, please just describe how to do the experiment.
  2. If “significantly” or “significant” was used (such as in line 210, 212, 213), it should be analyzed by statistical analysis and “P∠0.05” should be added in the text.
  3. “2.5 Non-specific immune response” changed to “immune response”

Author Response

Dear, reviewer
Thank you for your attentive review.
The revised content is included in the attached file.

Reviewer 3 Report

Dear Authors,

After carefully reviewing the authors’ response to my previous comments and suggestions and the new manuscript version, the issues have not been resolved, so the manuscript has not been sufficiently improved to warrant publication in Biology.

  1. The acclimatization period was modified from one week to one month, after my comment.
  2. As suggested by another previous reviewer the lethal dose should be determined.
  3. The issues have not been resolved, additional experiments needed.
  4. The issues have not been resolved, additional experiments needed.

Author Response

(The authors gave the same response as above.)

Reviewer 4 Report

  1. At line no. 17, write the word immune in small letters.
  2. At line no. 115, mark words 2, 7 in K2Cr2O7 as subscripts.
  3. Remove double parentheses in line no 116.
  4. At line no. 118, rewrite ‘was thoroughly exchanged once per two days and made the same concentration in the respective glass’ as ‘was thoroughly exchanged on alternate days, and the required concentration was again maintained in respective tanks.’
  5. From line no. 121-126, rewrite as: Liver and gill of the specimens were analyzed, and the actual amount of hexavalent chromium in each exposure experiment tank was measured and is shown in Table 2. Note: There is no need for intermediate lines between these sentences.
  6. Mention the complete form of ICP-MS in line no. 128.
  7. Remove sentences from line no. 131-139, as they are irrelevant in the methodology section.
  8. Mention full form of WST in line no. 152.
  9. Rewrite, ‘After all, processes, read the optical density at 405 nm’ as ‘After all this, optical density was measured at 450nm’ at line no. 186.
  10. Convert the sentence ‘After all, processes, read the optical density (O.D.) at 450 nm using a reader within 15 min after adding the stop solution’ at line no. 190 into past tense.
  11. Remove comma at line no. 258 between the words ROS production.
  12. Close parentheses at line no. 353 after reference 26.
  13. Some extra references have been mentioned in the reference section; please remove them.
  14. Add the following important references at appropriate places in the text
  • Farooq, S., Wali, A.F., Majid, S., Rasool, S., Wani, H.A., Bhat, S.A., Ali, S., Echikoti, R., Rasool, S., Ahmad, A. and Rehman, M.U., 2021. Neurotoxicity of Heavy Metals in Fishes: A Mechanistic Approach. In Freshwater Pollution and Aquatic Ecosystems (pp. 85-107). Apple Academic Press.
  • Firdous, A., Pillai, J.R., Rehman, M.U., Rashid, S.M., Rasool, S., Majid, S., Rashid, T., Farooq, A. and Masoodi, M.H., 2021. Toxicity of Heavy Metals in Freshwater Fishes: Challenges and Concerns. In Freshwater Pollution and Aquatic Ecosystems (pp. 25-51). Apple Academic Press.

Author Response

(The authors gave the same response as above.)

Round 2

Reviewer 3 Report

After carefully reviewing the authors' responses to my previous comments and suggestions and the new version of the manuscript, I think the authors have made the required changes.The manuscript still needs some corrections and improvement in the language which should be taken care of before final submission.

Author Response

Thank you for the reviewer's careful review.
This article received English corrections.